# Associations between Maternal and Infant Illness and the Risk of Postpartum Depression in Rural China: A Cross-Sectional Observational Study

**DOI:** 10.3390/ijerph17249489

**Published:** 2020-12-18

**Authors:** Wenbin Min, Wei Nie, Shuyi Song, Nan Wang, Weiqi Nie, Lanxi Peng, Zhuo Liu, Jingchun Nie, Jie Yang, Yonghong Ma, Yaojiang Shi

**Affiliations:** 1Center for Experimental Economics in Education, Shaanxi Normal University, Xi’an 710119, China; MinWenbin@snnu.edu.cn (W.M.); nieweiceee@163.com (W.N.); songshuyiceee@163.com (S.S.); wangnanceee@163.com (N.W.); nieweiqiceee@163.com (W.N.); penglanxiceee@163.com (L.P.); liuzhuoceee@163.com (Z.L.); jyang0716@163.com (J.Y.); shiyaojiang7@gmail.com (Y.S.); 2School of Public Health, Xi’an Medical University, Xi’an 710021, China; mayhong292152@163.com

**Keywords:** maternal and infant illness, risk of postpartum depression, maternal health knowledge, poor rural areas, observational study

## Abstract

This study explored how maternal and infant illness correlated with the risk of postpartum depression in the Chinese Qinba Mountains region. In total, 131 villages comprising 435 families with infants (≤6 months old) were randomly sampled. We collected data on maternal and infant illnesses and maternal health knowledge level. The Depression, Anxiety, and Stress Scale-21 was used to measure the risk of postpartum depression. We used descriptive statistics and multivariate logistic regression for the analysis. Infant overall health status was a risk factor for postpartum depression (odds ratio (OR) = 1.90, 95% Confidence Interval (95% CI) = 1.10~3.28), whereas maternal overall health status was not correlated with postpartum depression (OR = 1.36, 95% CI = 0.55~3.39). For specific illnesses, infants experiencing over two common illnesses in the past two weeks (OR = 1.98, 95% CI = 1.13~3.45) and mothers experiencing over two common pains within two weeks after delivery (OR = 1.77, 95% CI = 1.02~3.08) were risk factors for postpartum depression, whereas infants with mild and severe stunted growth, maternal C-section, and postpartum body mass index (normal or overweight) were not correlated with it (all *p* > 0.050). Maternal health knowledge was an important moderator of maternal and infant illnesses on the risk of postpartum depression. In conclusion, maternal and infant illness were essential factors for the risk of postpartum depression in a poor rural region in western China, which may be mainly affected by the feeling of uncertainty of illness. Improved maternal and infant health and enhanced maternal health knowledge might alleviate the risk of postpartum depression.

## 1. Introduction

Postpartum depression affects women worldwide. Presently, this condition is treated by providing psychological counseling and social support [1]. The World Health Organization (WHO) indicates that the first 6 months after childbirth are crucial for the development of postpartum depression; moreover, the health status of mothers and infants is closely related to the risk of postpartum depression [2,3]. Compared with urban areas, poor rural areas of China have a lower quality of medical services, as well as lower levels of income and maternal education. Thus, mothers and infants may experience more illnesses throughout their lives, such as stunted child growth, anemia, and maternal reproductive illness after childbirth [4,5]. Additionally, mothers in poor rural areas in China have relatively low levels of knowledge about nutrition and health. For example, the study of Huo et al. from poor areas showed that the infant caregivers’ nutritional knowledge awareness rate was only 29.6% while this rate was 40.3% in urban areas [6,7]. Maternal and infant illnesses may affect the mother’s postpartum mental state in two ways: physically, through pain and hormone level alterations caused directly by illness [8], and psychologically, through negative events and uncertainty related to illness. This uncertainty manifests as an individual’s lack of sufficient knowledge and experience to effectively recognize and understand the meaning and consequences of an event [9]. It leads to psychological pressure and anxiety, affecting the mother’s psychological status [10].

Previous studies have focused on the influence of psychological counseling, social support, and family care on the risk of postpartum depression [1]. However, maternal and infant illnesses may represent important risk factors for postpartum depression, potentially providing a new strategy to lower the incidence of the risk of postpartum depression through improving the health of mothers and infants. Therefore, this study aimed to determine the association between maternal and infant illness and the risk of postpartum depression in a population from a poor rural region in western China.

## 2. Materials and Methods

### 2.1. Study Population and Sampling

We conducted a population-based cross-sectional study on pregnant women and infants (≤6 months old) in rural areas of the Qinba Mountains region from March to April 2019. According to previous literature, assuming a sample mean of 0.20, a sampling error of 0.05, a standard deviation of 0.25, a significance level of 0.05, and a response rate of 10%, a research sample of at least 450 is required. The sampling process was as follows: first, we selected 13 national-level impoverished counties in two prefectures of the Qinba Mountains region to include in the sample. Subsequently, we obtained a list of villages in each county and determined the total number of pregnant women and families with infants in each village. Considering the survey cost and organization, villages with small (<3 samples) and large (>15 samples) sample sizes were excluded. Based on power calculations, 131 villages were randomly selected from all eligible villages and all pregnant women and families with infants were included as participants. In total, 495 eligible participants were enrolled in the study. Of these, 28 participants were excluded because mothers were not the first caregivers of infants and 32 were excluded because of missing data. Finally, 435 participants were included in the analysis (Figure 1).

### 2.2. Measurements

We used questionnaires to collect information on general demographic characteristics, including maternal (age and education level), infant (sex, age, and number of siblings), and family (household income and infant’s second caregiver) characteristics.

According to the health status of mothers and infants and the WHO guidelines [11,12,13,14,15,16], we collected information on maternal and infant illness. 

Regarding infant illness, first we collected information on whether the infant has had more than two common illnesses in the past two weeks (such as fever, diarrhea, cough, and cut/burn) and their Length-for-Age Z-Score (LAZ) to assess their development. The infant’s length was measured by a local nurse using a professional scale for infants. The LAZ score was calculated according to the WHO infant growth and development measurement standards: a score between −3 and −2 represented mild stunted growth, while a score lower than −3 represented severe stunted growth. Secondly, according to the occurrence of the above infant illnesses, the infant’s overall health status was dichotomized: if they had any of the above conditions, the overall health status was considered present (1), meaning infants having poor overall health status; otherwise, they were considered absent (0), meaning infants having better overall health status.

Regarding postpartum illness, firstly, we collected information on whether the mother had more than two common pains within two weeks after delivery (such as breast pain, back pain, or breast infection), whether the delivery was through a cesarean section, and whether the mother had an abnormal body mass index (BMI). A nurse measured the weight (kg, in kilograms) and height (m, in meters) of the mothers and the project team calculated the corresponding BMI according to the following formula: BMI = weight (kg)/(height (m))^2^. A BMI lower than 19.8 (underweight) or greater than 25 (overweight) was considered abnormal. Secondly, according to the occurrence of the above maternal illnesses, the maternal overall health status was dichotomized: if they had any of the above conditions, postpartum overall health status was considered present (1), meaning mothers having poor overall health status; otherwise, they were considered absent (0), meaning mothers having better overall health status.

A self-reported questionnaire developed according to the feeding guideline for Chinese infants was used to measure the level of maternal and infant health knowledge of mothers [17,18], as shown in the Appendix A. The questionnaire comprised 9 questions, including infant cold prevention, breastfeeding, and methods of mental adaptation after childbirth. Each correct answer was assigned 1 point, with a possible total score ranging from 0 to 9 points. A score of 5 or greater was considered a good level of knowledge (1), while a score of 4 or less was considered poor (0).

The Depression, Anxiety, and Stress Scale-21 (DASS-21) contains 21 items divided into three subscales for depression, anxiety, and stress [19]. This scale was effective for the Chinese mainland population and was used in the investigation and screening of the risk of postpartum depression in rural China [20,21,22,23,24,25], with subscales’ internal consistency index (Cronbach’s alpha) ranging from 0.754 to 0.823, an overall internal consistency index of 0.912, and a one-week test–retest reliability of 0.751 [20]. We used only the depression subscale because the focus of this study was on the risk of postpartum depression. The depression subscale’s internal consistency index in this study was 0.777, which is consistent with that of previous studies [20,25]. Each of the 7 items in the depression subscale has four possible responses: “Did not apply to me at all” (0 point), “Applied to me to some degree or some of the time” (1 point), “Applied to me to a considerable degree or a good part of the time” (2 points), and “Applied to me very much or most of the time” (3 points). The score of each question is multiplied by 2 to obtain the risk of postpartum depression score, resulting in a total score ranging from 0 to 42 points. In accordance with previous studies, a score greater than or equal to 10 points was considered positive for the risk of postpartum depression, while a score lower than 10 points was considered negative [21,22,23].

### 2.3. Data Collection

This study adopted the questionnaire survey method, which was completed through in-person interviews. Before the interviews, all investigators were uniformly trained and participated in pre-survey exercises to ensure accuracy and consistency when conducting the interview. The investigators conducted door-to-door interviews with eligible mothers, informing them in detail regarding the significance, purpose, and operation methods of the study, as well as the cooperation required for participation in it, before the start of the survey. After obtaining the mothers’ informed consent and cooperation, investigators collected as much real information as possible. Other family members were not allowed to be present during the interview to ensure the authenticity of the questionnaire. Project team members were present during the survey as leading interviewers. They were trained to provide standardized instructions and answer questions regarding the questionnaire items in a neutral way to avoid interference. For those with less education or difficulty in understanding, the questionnaire was completed with the help of the examiners. The height and weight of the mothers were measured as part of the interview by professional nurses.

### 2.4. Statistical Analysis

STATA 15.1 (StataCorp, College Station, TX, USA) was used for the statistical analyses. We performed a descriptive statistical analysis of the risk of postpartum depression using frequency, percentage, mean and standard deviation as appropriate. Differences between categorical data were analyzed using Pearson’s chi-squared test. Multivariate logistic regression analysis was used to explore the associations between the risk of postpartum depression and maternal and infant overall health status as well as different types of illness. *p* < 0.050 was considered statistically significant.

### 2.5. Ethical Approval

We followed the principles of the Declaration of Helsinki and received ethical approval from the Institutional Review Board of Shaanxi Normal University (Xi’an, China) and Stanford University (PANW, USA, No.44312). Permission was received from every participant before the interview. 

## 3. Results

### 3.1. The Risk of Postpartum Depression

Of the 435 mothers included in this study, 68 (15.6%) were at risk of postpartum depression, whereas 367 (84.4%) were not. Their average age was 28 ± 5 years, and 339 mothers (77.9%) had junior high school-level education or above. Regarding the infants, 234 (53.8%) were male, the average age was 3.6 months, and 164 (37.7%) were only children. Overall, 219 infants (50.3%) had poor overall health status. Of them, 177 (40.7%) had more than two common illness symptoms over the past two weeks, 40 (9.2%) had their growth mildly stunted, and 21 (4.8%) had their growth severely stunted. Simultaneously, 380 mothers (87.4%) had poor postpartum overall health status. Of them, 216 (49.7%) had two or more common pain symptoms two weeks after delivery, 193 (44.4%) had a cesarean section, and 242 (55.6%) had abnormal BMI values.

Compared with mothers without the risk of postpartum depression, those with the risk of postpartum depression had more infants with poor overall health status (*t* = −2.32, *p* = 0.021) and more infants that had more than two common illness symptoms over the past two weeks (*t* = −2.80, *p* = 0.005). They also differed significantly in the occurrence of common pain symptoms two weeks after delivery (*t* = −2.18, *p* = 0.030) and in the level of maternal health knowledge (*t* = 3.28, *p* = 0.001) (Table 1).

### 3.2. Association between Maternal and Infant Illnesses and the Risk of Postpartum Depression

Logistic regression analysis was performed to determine whether the risk of postpartum depression was the outcome variable and whether overall health status of mothers and infants were independent variables. The presence of mothers having poor overall health status after childbirth was not associated with the risk of postpartum depression, whereas the presence of having poor overall health status in infants was positively associated with the risk of postpartum depression (odds ratio (OR) = 1.90, 95% Confidence Interval (95% CI) = 1.10~3.28). Considering marginal effects, infants having poor overall health status may contribute directly to 45.5% (7.1%/15.6%) of the risk of postpartum depression occurrence in this population (Table 2).

Additionally, for specific illnesses, logistic regression showed that the occurrence of more than two common illnesses symptoms in infants over the past two weeks was positively correlated with the risk of postpartum depression (OR = 1.98, 95% CI = 1.13~3.45). Contrastingly, mild or severe stunted growth was not correlated with the risk of postpartum depression. Regarding maternal specific illnesses, the occurrence of more than two common pains within two weeks after delivery was positively correlated with the risk of postpartum depression (OR = 1.77, 95% CI = 1.02~3.08). Mothers who underwent a cesarean section and had a postpartum BMI that was either underweight or overweight were not associated with risk of postpartum depression (Table 2).

### 3.3. Association for Subgroup of Maternal Health Knowledge Level

Logistic regression was performed in subgroups according to maternal health knowledge levels. When the mother had a poor health knowledge level, the infant’s overall health status was positively correlated with the risk of postpartum depression (OR = 2.69, 95% CI *=* 1.27~5.69). In contrast, when the mother had a good health knowledge level, the infant’s overall health status was not correlated with the risk of postpartum depression. Additionally, in both of the subgroups, the maternal overall health status was not associated with the risk of postpartum depression (Table 3).

## 4. Discussion

Our survey including poor rural areas of western China identified that 15.6% of mothers experience a risk of postpartum depression. This result is similar to previous findings which found that the rate of the risk of postpartum depression in poor rural areas of western China was higher than the 10.9~14.8% reported in the meta-analysis in populations of urban and developed eastern areas [26,27]. Postpartum depression affects the mother, their family relationships, parenting ability, and their child’s cognitive, social, and emotional development [28,29]. Simultaneously, the health status of mothers and infants in rural China is worse than that of those residing in urban areas, with higher inflammation and infant developmental delay rates [30,31]. 

Our results showed a positive correlation between the infant’s overall health status and risk of postpartum depression, whereas no correlation was found between maternal overall health status and risk of postpartum depression. Previous studies showed that although most infant illnesses did not pose a serious risk to infants’ lives, caring for sick infants brought an additional physical and psychological burden to mothers. It may be related to the increased risk of postpartum depression [1,32]. At the same time, maternal and infant illnesses have influenced maternal mental health after delivery differently [33,34]. Maternal illness might affect the mother’s mental health physically (pain and hormonal alterations caused by illness) and psychologically (e.g., uncertainty caused by negative events) [8,34,35]. While the infant’s illness cannot cause direct physical pain to the mother, it may have a psychological impact. According to the theory of illness uncertainty, the mother’s sense of uncertainty was lessened when the location, severity, symptoms patterns, and frequency of the illness are known [34,35]. Due to the difficulty of determining the cause, specific location, and pain degree of an infant who was unable to communicate, the sense of illness uncertainty was heightened, significantly impacting the risk of postpartum depression [10,33].

We found that the occurrence of symptoms of common illnesses in infants over the past two weeks and of symptoms of common pains in mothers during the first two weeks after childbirth correlated with maternal psychological status, while infant stunted growth, cesarean section, and BMI after childbirth did not. Previous studies have shown that fever, diarrhea, cough, and cut/burns and breast pain, back pain, and breast infections represent infant and maternal acute illnesses, respectively [8,11]. Contrastingly, infant stunted growth, cesarean section, and abnormal BMI after childbirth represent chronic illnesses [36,37,38,39]. The pain degree, incidence frequency, and duration of maternal and infant acute illnesses had no obvious regularity [40,41,42,43], whereas the growth and developmental status of infants, maternal cesarean section, under- or overweight BMI after childbirth, and other chronic maternal and infant illnesses had a slower progress and more recognizable symptoms [44,45]. Moreover, the treatment process of maternal and infant acute illnesses was relatively complex, with a less predictable outcome and higher possibility of sudden deterioration during treatment [11,42,46]. For example, acute diarrhea could lead to infant death within 2 h [47]. Therefore, compared with chronic illness, maternal and infant acute illnesses might cause a more acute sense of uncertainty. Our results also indicated that maternal and infant acute illnesses with a higher degree of uncertainty were more likely to correlate with the risk of postpartum depression, whereas mild and severe stunted growth did not. It suggested that if the uncertainty caused by the illness was relatively low, infant illness would not correlate with the risk of postpartum depression, even if the illness could be considered severe.

Our results also indicated that the level of maternal health knowledge played an important regulatory role in the correlation between maternal and infant illness and postpartum mental health. The mothers’ perceptions of illness were affected by their knowledge level, cognitive structure, and other factors. When their level of maternal health knowledge was low, their understanding of the illness and treatment was negatively impacted, and they were less able to estimate its possible consequences [48,49]. Therefore, the same illness might cause different levels of uncertainty based on the mother’s health knowledge level. Moreover, a good health knowledge level could improve the mother’s ability to manage the symptoms of common illnesses, affecting the illness treatment, severity degree, pain intensity, and frequency of monitoring [9,50], consequently reducing the uncertainty caused by the illness and, to a certain extent, the likelihood of developing the risk of postpartum depression. Therefore, strategies to improve the health education of mothers, for example, by trained community health workers, are necessary to allow them to appropriately recognize and manage maternal and infant illness, as well as infant behavior, in the early postpartum period [51].

Studies have also shown that the risk of postpartum depression may have a reverse causal relationship with maternal and infant health. For example, the risk of postpartum depression in mothers was associated with a higher incidence of respiratory infections, diarrhea, and stunted growth in infants [52,53]. However, no study has shown that the risk of postpartum depression was associated with an increased occurrence of common pain symptoms two weeks after childbirth. Therefore, the correlation between maternal and infant illnesses and the risk of postpartum depression could not be fully explained by the negative effects of the risk of postpartum depression, and the sense of uncertainty brought by maternal and infant illness was still an important factor to understand this correlation.

Therefore, to reduce the risk of postpartum depression in mothers residing in poor rural areas of western China, maternal and infant health status should be closely monitored, in addition to providing psychological counseling or direct interventions. Contrastingly, compared with that of residents of urban areas, adverse maternal and infant overall health status occurs more frequently in residents of poor rural areas of western China. Therefore, maternal and infant public health services in this region must be improved to positively impact their health status.

This study has some limitations: first, the cross-sectional survey study design impacts the ability to determine causal relationships between maternal and infant illness and the risk of postpartum depression; and second, the uncertainty caused by maternal and infant illness was not directly measured. Future studies should include measurements of illness uncertainty of illness using relevant scales (e.g., the Parental Perception of Uncertainty Scale) to directly measure the relationship between uncertainty caused by maternal and infant illness and the risk of postpartum depression, improving the level of discussion regarding this condition.

## 5. Conclusions

In summary, this study showed that maternal and infant illness was associated with the risk of postpartum depression in western rural China. Mothers’ health knowledge level was an important moderator of the effect of maternal and infant illnesses on the risk of postpartum depression. Our findings implied that maternal and infant illness might affect the risk of postpartum depression through feelings of uncertainty of illness. Improved maternal and infant health and enhanced maternal health knowledge might alleviate the risk of postpartum depression.

## Figures and Tables

**Figure 1 ijerph-17-09489-f001:**
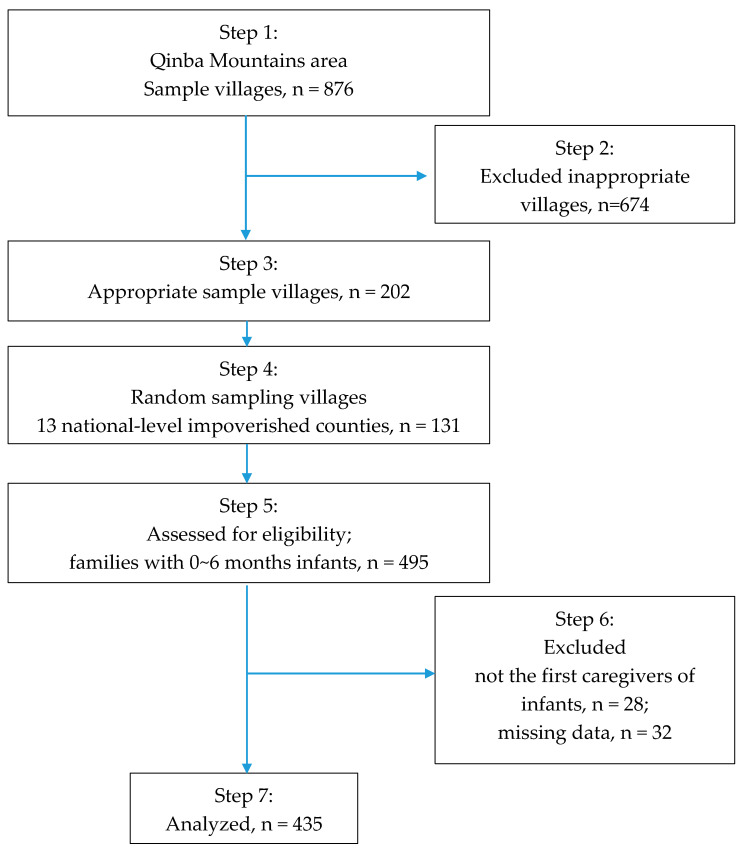
Flow diagram of sampling.

**Table 1 ijerph-17-09489-t001:** Risk of postpartum depression of mothers in different groups of maternal and infant illness status and the level of maternal health knowledge.

Variable		Total Sample		Non-Risk of Postpartum Depressive DASS < 10		Risk of Postpartum Depressive DASS ≥ 10		Non-risk of Postpartum Depression vs. Risk of Depression	
		Number	%	Number	%	Number	%	*t*	*p*
**Infan** **t**									
**Infant overall health status**								−2.32	**0.021**
	No	216	49.7	191	88.4	25	11.6		
	Yes	219	50.3	176	80.4	43	19.6		
**More than two common illnesses**								−2.80	**0.005**
	No	258	59.3	228	88.4	30	11.6		
	Yes	177	40.7	139	78.5	38	21.5		
**Mild** **stunted**								−0.11	0.908
	No	395	90.8	333	84.3	62	15.7		
	Yes	40	9.2	34	85.0	6	15.0		
**Severe** **stunted**								−0.44	0.660
	No	414	95.2	350	84.5	64	15.5		
	Yes	21	4.8	17	81.0	4	19.0		
**Maternal**									
**Maternal overall health status**								−1.03	0.302
	No	55	12.6	49	89.1	6	10.9		
	Yes	380	87.4	318	83.7	62	16.3		
**More than two common pains**								−2.18	**0.030**
	No	219	50.3	193	88.1	26	11.9		
	Yes	216	49.7	174	80.6	42	19.4		
**Cesarean section**								−0.22	0.825
	No	242	55.6	205	84.7	37	15.3		
	Yes	193	44.4	162	83.9	31	16.1		
**BMI**								−1.11	0.268
	Thin or overweight	242	55.6	200	82.6	42	17.4		
	Normal	193	44.4	167	86.5	26	13.5		
**Maternal health knowledge**								3.28	**0.001**
	Poor	217	49.9	172	79.3	45	20.7		
	Good	218	50.1	195	89.4	23	10.6		

Note: Values in bold are significant at the *p* < 0.050 level.

**Table 2 ijerph-17-09489-t002:** Logistic regression on the correlation between maternal and infant illness status and the risk of postpartum depression.

Variable		Risk of Postpartum Depression					
		*p*	OR	95% CI	*p*	OR	95% CI
**Infant and maternal**							
	Infant overall health status	**0.021**	1.90	1.10~3.28			
	Maternal overall health status	0.506	1.36	0.55~3.39			
**Infant illness**							
	More than two common illnesses				**0.016**	1.98	1.13~3.45
	Mild stunted				0.745	1.17	0.45~3.08
	Severe stunted				0.485	1.52	0.47~4.97
**Maternal illness**							
	More than two common pains				**0.044**	1.77	1.02~3.08
	Cesarean section				0.849	0.95	0.55~1.64
	BMI thin or overweight				0.282	1.36	0.78~2.37
**Maternal characteristics**							
	Age	0.535	1.02	0.96~1.08	0.458	1.02	0.96~1.09
	Education > Junior high school	0.289	0.70	0.36~1.36	0.306	0.70	0.36~1.38
**Infant characteristics**							
	Male	0.546	1.18	0.69~2.03	0.587	1.16	0.67~2.01
	1~2 months	0.849	0.93	0.46~1.88	0.794	0.91	0.44~1.88
	3~4 months	0.437	1.28	0.69~2.39	0.377	1.33	0.71~2.51
	Siblings	0.629	1.13	0.69~1.85	0.639	1.13	0.68~1.86
**Family characteristics**							
	Income > 30,000 CNY	0.783	0.93	0.54~1.60	0.729	0.91	0.52~1.58
	Second caregiver (infant’s grandmother)	0.330	0.58	0.19~1.74	0.405	0.62	0.20~1.91
	Second caregiver (infant’s father)	0.671	0.78	0.25~2.48	0.798	0.86	0.26~2.79

Note: The city fixed effect is controlled. Values in bold are significant at the *p* < 0.050 level.

**Table 3 ijerph-17-09489-t003:** Logistic regression on the correlation in two subgroups of maternal health knowledge.

Variable		Risk of Postpartum Depression		
		*p*	OR	95% CI
**Maternal health knowledge: poor**				
	Infant overall health status	**0.010**	2.69	1.27~5.69
	Maternal overall health status	0.250	2.51	0.52~12.01
**Maternal health knowledge: good**				
	Infant overall health status	0.481	1.40	0.55~3.55
	Maternal overall health status	0.483	0.63	0.18~2.27

Note: The city fixed effect is controlled. Due to space reasons, the control variable coefficients (including Maternal characteristics: Age, Education > Junior high school; Infant characteristics: Male, 1~2 months, 3~4 months, Siblings; Family characteristics: Income > 30,000 CNY, Second caregiver (infant’s grandmother), Second caregiver (infant’s father)) are not shown again. Value in bold is significant at the *p* < 0.050 level.

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
