# Peer review of "Associations between Maternal and Infant Illness and the Risk of Postpartum Depression in Rural China: A Cross-Sectional Observational Study"

_ijerph, 2020, doi:10.3390/ijerph17249489_

Round 1
Reviewer 1 Report
The article is well-written. Following are my comments:
- Line 83: BMI =m/h2 needs to be corrected. It is weight (kg) / (height (m))2
- A flow diagram of the methods and data collection techniques followed can be incorporated for easy visual to the readers.
- Minor checks are required for typographical accuracy.
- The limitations mentioned in the conclusion section may be removed from there and inserted in the discussion section.
Reviewer 2 Report
This is a study on a few overlooked risk factors of postpartum depression in women living in rural China. I found the topic interesting and the data collection impressive. But the organization of the manuscript and the presentation of the results need improvement. I attach a pdf file with some specific suggestions, which i hope would be helpful for the authors to understand what their readers are looking for. Here are my general comments.
Abstract:
Please be clear and consistent in the results. Sentences in line 18 to 20 contradict with each other. Also, the conclusion and recommendation is not strongly supported by the finding. I suggest be very cautious in making such recommendation.
Background:
The authors could have give some more background on maternal health in rural Chinese population. I was wondering what was the knowledge gap.
Methods:
I found the sampling clear, but the description of the variable definition, classification, and modelling very unclear. This hindered my understanding of the tables.
Results:
I suggest presenting OR and 95%CI for logistic regression. All those listed statistics made it very hard for me to see the results
Discussion:
Organization was the problem. I expected to read a summary of the finding in the 1st paragraph, some discussion relating to previous studies, and interpretation based on theories, then limitations and strengths. The conclusion should be brief and on the point, no extrapolation from the finding.

Reviewer 3 Report
This is an interesting paper that requires a little more explanation in some areas and some correction of English is others.
Comments: Last sentence of Abstract - I would write the sentence the other way around.
Key words: not sure "poverty-stricken rural areas" is an appropriate keyword - perhaps just rural area and poverty separately?
Need more info/detail on what power calculation you used when sampling the villages.
I think it would be good to include (perhaps as an appendix) the 8 questions asked about maternal knowledge as interesting to the reader.
I think most scales that ask questions about depression are not clinically applied rather the scores indicate depression and as such are not a diagnosis and so the language around that may need to be clarified throughout the paper.
Results: line 135 - add the (%) after each n (164) for consistency.
Line 136-137 had their growth mildly dysplasia - not correct grammar
Table 1&2: has there been any multiple testing correction applied to these data as otherwise could be giving spurious significance.
All tables - I would bold the significant P values for readability.
Line 165 - A BMI - not correct grammar.
Discussion - several sentences need English grammar checked and corrected i.e. line 188, 191, 193, 197 (Because), 216, 217.
I would also be careful in the Discussion as this is only an association study not to overstate your findings.
Reviewer 4 Report
Your research was interesting and impressive.
In the results, How about moving the first variable in Table 1,'infant health status', to the title?
Similarly, consider moving the first variable in Table 2 to the title.
In the discussion, you mentioned that 'The diseases can cause different levels of uncertainty depending on the mother's level of health knowledge', so I look forward to discussing the policy implications.
Round 2
Reviewer 2 Report
Thanks to the author for clarifying. I have no further comments.
Author Response
Thank you again for your comments! Best wishes!